# Multi-layer State Evolution Under Random Convolutional Design

**Max Daniels**
Northeastern University
daniels.g@northeastern.edu

**Cédric Gerbelot**
ENS Paris
cedric.gerbelot@ens.fr

**Florent Krzakala**
IdePHIcs Laboratory, EPFL
florent.krzakala@epfl.ch

**Lenka Zdeborová**
SPOC Laboratory, EPFL
lenka.zdeborova@epfl.ch

## Abstract

Signal recovery under generative neural network priors has emerged as a promising direction in statistical inference and computational imaging. Theoretical analysis of reconstruction algorithms under generative priors is, however, challenging. For generative priors with fully connected layers and Gaussian i.i.d. weights, this was achieved by the multi-layer approximate message (ML-AMP) algorithm via a rigorous state evolution. However, practical generative priors are typically convolutional, allowing for computational benefits and inductive biases, and so the Gaussian i.i.d. weight assumption is very limiting. In this paper, we overcome this limitation and establish the state evolution of ML-AMP for random convolutional layers. We prove in particular that random convolutional layers belong to the same universality class as Gaussian matrices. Our proof technique is of an independent interest as it establishes a mapping between convolutional matrices and spatially coupled sensing matrices used in coding theory.

## 1 Introduction

In a typical signal recovery problem, one seeks to recover a data signal $x_0$ given access to measurements $y_0 = G_\theta(x_0)$, where the parameters $\theta$ of the signal model are known. In many problems, it is natural to view the measurement generation process as a composition of simple forward operators, or 'layers.' In this work, we are concerned with multi-layer signal models of the form

$$G_\theta(x_0) = \varphi^{(1)}(W^{(1)}\varphi^{(2)}(W^{(2)}\ldots\varphi^{(L)}(W^{(L)}x_0;\zeta^{(L)})\ldots). \tag{1}$$

where $W^{(l)} \in \mathbb{R}^{n_{l-1} \times n_l}$ are linear sensing matrices and where $\varphi^{(l)}(z;\zeta)$ are separable channel functions that may be non-linear and may depend on unknown channel noise $\zeta$. In the multi-layer case $L > 1$, this signal model can be viewed as a simple instance of recovery under a Generative Neural Network (GNN) prior, a technique which has recently shown promise as a generalization of sparsity priors for signal processing applications Bora et al. [2017]. For example, Gaussian compressive sensing $y_0 = AG_\theta(x_0) + \zeta$ under a prior with random Gaussian weights can be naturally viewed as an instance of the multilayer signal model, and given an estimate $\hat{x}$, one can recover the target signal $\hat{s} = G_\theta(\hat{x})$ from the weights $\{W^{(l)}\}$ and noise-independent channel functions $\{\varphi^{(l)}(z)\}$.

In practical settings GNNs often use structured *convolutional weight matrices*, preventing the direct application of estimators that require $W^{(l)}$ to have unstructured entries. Motivated by this, we take interest in a variant of the recovery problem (1) in which some of the sensing matrices $W^{(l)}$ may be

36th Conference on Neural Information Processing Systems (NeurIPS 2022).

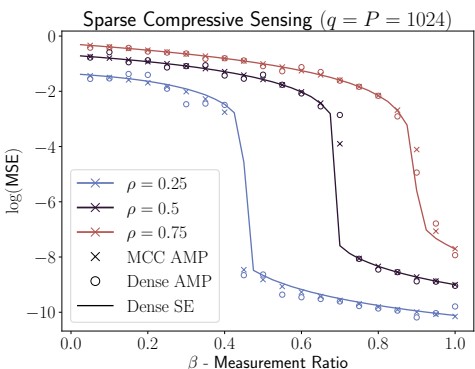 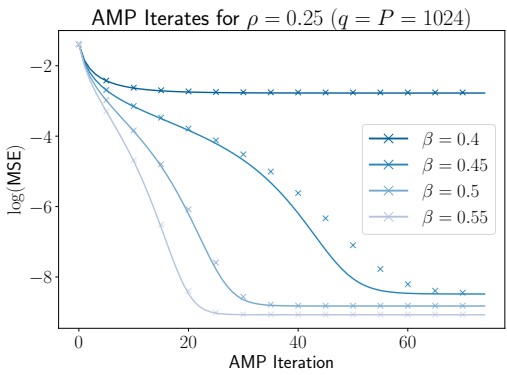

Figure 1: Agreement between the performance of the AMP algorithm run with random multichannel convolutional matrices and its state evolution as proven in this paper. (**left**) Compressive sensing $y_0 = W x_0 + \zeta$ for noise $\zeta_i \sim \mathcal{N}(0, 10^{-4})$ and signal prior $x_0 \sim \rho \mathcal{N}(0,1) + (1-\rho)\delta(x)$, where $W \in \mathbb{R}^{Dq \times Pq}$ has varying aspect ratio $\beta = D/P$. Crosses correspond to AMP evaluations for $W \sim \text{MCC}(D, P, q, k)$ according to Definition 3.2, averaged over 10 independent trials. Dots correspond to AMP evaluations for $W \in \mathbb{R}^{D \times P}$ with i.i.d. Gaussian entries $W_{ij} \sim \mathcal{N}(0, 1/P)$. Lines show the state evolution predictions when $W_{ij} \sim \mathcal{N}(0, 1/Pq)$. The system size is $P = 1024$, $q = 1024$, $k = 3$, where $\beta$ and $D = \beta P$ vary. While our theorem treats the limit $P, D \to \infty$, $q, k = O(1)$, we observe strong empirical agreement even when $q \sim P$. In Appendix C.1 we give the same figure for $q = 10 \ll P$. (**right**) AMP iterates at $\rho = 0.25$ and $\beta$ near the recovery transition. Rather than showing these models have equivalent fixed points, we show a stronger result: the state evolution equations are equivalent *at each iteration*.

*multi-channel convolutional* (MCC) matrices, having a certain block-sparse circulant structure which captures the convolutional layers used by many modern generative neural network architectures Karras et al. [2018, 2019].

In this work, we develop an asymptotic analysis of the performance of an *Approximate Message Passing* (AMP) algorithm Donoho et al. [2009] for recovery from multichannel convolutional signal models. This family of algorithms originates in statistical physics Mézard and Montanari [2009], Zdeborová and Krzakala [2016] and allows to compute the marginals of an elaborate posterior distribution defined by inference problems involving dense random matrices. A number of AMP iterations have been proposed for various inference problems, such as compressed sensing Donoho et al. [2009], low-rank matrix recovery Rangan and Fletcher [2012] or generalized linear modeling Rangan [2011]. More recently, composite AMP iterations (ML-AMP) have been proposed to study multilayer inference problems Manoel et al. [2017], Aubin et al. [2019]. Here we consider the ML-AMP proposed in Manoel et al. [2017] to compute marginals of a multilayer generalized linear model with unstructured random weights, but we replace these weights random convolutional weight matrices. A major benefit of AMP lies in the fact that the asymptotic distribution of their iterates can be exactly determined by a low-dimensional recursion: the state evolution equations. This enables to obtain precise theoretical results for the reconstruction performance of the proposed algorithm. Another benefit of such iterations is their low computational complexity, as they only involve matrix-multiplication and, in the separable case, pointwise non-linearities.

Previous works on AMP suggest that the state evolution is not readily applicable to our setting because its derivation requires strong independence assumptions on the coordinates of the $\{W^{(l)}\}$ which are violated by structured multi-channel convolution matrices. Despite this, we use AMP for our setting and rigorously prove its state evolution. Our main contributions are:

1. We rigorously prove state evolution equations for models of the form (1), where weights are allowed to be either i.i.d. Gaussian or random structured MCC matrices, as in Definition 3.2.

2. For separable channel functions $\varphi^{(l)}$ and separable signal priors, we show that the original ML-AMP of Manoel et al. [2017] used with sensing matrices that are either dense Gaussian matrices or random convolutional ones admits the same state evolution equations, up to a rescaling. Multi-layer MCC signal models can therefore simulate dense signal models while making use of fast structured matrix operations for convolutions.

3. The core of our proof shows how an AMP iteration involving random convolutional matrices may be reduced to another one with dense Gaussian matrices. We first show that random convolutional matrices are equivalent, through permutation matrices, to dense Gaussian ones with a (sparse) block-circulant structure. We then show how the block-circulant structure can be embedded in a new, matrix-valued, multilayer AMP with dense Gaussian matrices, the state evolution equations of which are proven using the results of Gerbelot and Berthier [2021], with techniques involving spatially coupled matrices Krzakala et al. [2012], Javanmard and Montanari [2013].

4. We validate our theory numerically and observe close agreement between convolutional AMP iterations and its state evolution predictions, as shown in Figure 1 and in Section 5. Our code can be used as a general purpose library to build compositional models and evaluate AMP and its state evolution. We make this code available at `https://github.com/mdnls/conv-ml-amp.git`.

## 2 Related Work

AMP-type algorithms arose independently in the contexts of signal recovery and spin-glass theory. In the former case, Donoho et al. [2009] derives AMP for Gaussian compressive sensing. This approach was later generalized by Rangan [2011] to recovery problems with componentwise *channel functions* that may be stochastic and/or nonlinear, and generalized further by Manoel et al. [2017] to multi-layer or compositional models. Due to the versatility of this approach, a wide variety of general purpose frameworks for designing AMP variants have since been popularized Fletcher et al. [2018], Baker et al. [2020], Gerbelot and Berthier [2021]. Proof techniques to show the concentration of AMP iterates to the state evolution prediction developed alongside new variants of the algorithm. In the context of spin-glass theory, Bolthausen's seminal work Bolthausen [2009] introduces a Gaussian conditioning technique used widely to prove AMP concentration. Following this approach, Bayati and Montanari [2011], Javanmard and Montanari [2013], Berthier et al. [2020] treat signal models with dense couplings and generalized channel functions. More recently, a proof framework adaptable to composite inference problems was proposed in Gerbelot and Berthier [2021], which we use in our proof.

There has also been significant interest in relaxing the strong independence assumptions required by AMP algorithms on sensing matrix coordinates. In one direction, *Vector AMP* (VAMP) algorithms target signal models whose sensing matrices are drawn from *right orthogonally invariant* distributions. The development of VAMP algorithms followed a similar trajectory to that of vanilla AMP Schniter et al. [2016], Fletcher et al. [2018], Rangan et al. [2019], Baker et al. [2020]. The MCC ensemble considered in this work is not right orthogonally invariant, but we observe strong empirical evidence that an analogous version of Theorem 4.2 holds for VAMP as well, as described in Appendix C.2. In a second direction, there has been much interest in *spatial coupling* sensing matrices, which were used to achieve the information-theoretically optimal performance in sparse compressive sensing Donoho et al. [2013], Barbier et al. [2015], Krzakala et al. [2012], with complementary state evolution guarantees Javanmard and Montanari [2013]. The concept of spatial coupling and proofs of its performance originated in the literature of error correcting codes Kudekar et al. [2011, 2013], where it developed from the so-called convolutional codes Felstrom and Zigangirov [1999]. The connection between spatial coupling and convolution layers of neural networks, that we establish in this paper, is as far as we know novel.

Another direction of related work is the design of generative neural network architectures, and correspondingly, the design of signal recovery procedures that can make use of new generative prior models. Bora et al. [2017], one of the original works on compressive sensing with generative models, studies signal recovery under a fully connected VAE prior and a convolutional DC-GAN prior. More advanced convolutional architectures such as PG-GAN Karras et al. [2018] and Style-GAN Karras et al. [2019] have been studied in followup work as priors for a variety of signal recovery problems Daras et al. [2021], Gu et al. [2020]. For simplicity and theoretical tractibility, we do not consider some fine-grained practical modifications used by these architetures, like batch normalization Ioffe and Szegedy [2015] or strided convolution layers Radford et al. [2015], focusing instead on the essential elements of simple convolutional networks. Lastly, while our focus is on feedforward convolutional priors such as GAN/VAE networks, there is growing interest in alternative approaches to signal recovery under neural network priors, such as normalizing flows Rezende and Mohamed

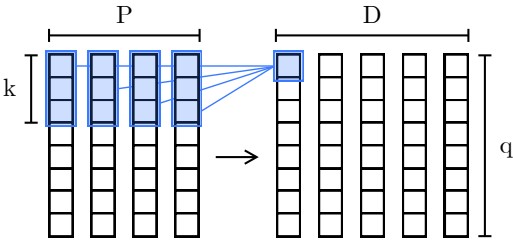

Figure 2: MCC matrices operate on $Pq$ dimensional input data, composed of $q$-dimensional signals for each of $P$ separate channels. The $i$-th output channel is a linear combination of convolutional features extracted from input channels, where $k$ is the convolutional filter size: $y^{(i)} = \sum_{j=1...P} C_{ij} x^{(j)}$. Each row of $W$ is a concatenation of $P$ different rows from convolutional matrices, each row hvaing $k$ nonzero entries, for a total of $kP$ nonzero entries. Blue boxes show which $kP$ input entries contribute to a single output entry.

[2015], Dinh et al. [2015, 2017], Kingma and Dhariwal [2018], Asim et al. [2020] and score-based generative models Song and Ermon [2020], Jalal et al. [2021]. These approaches fall outside the scope of our work and may be interesting directions for future investigation.

## 3    Definition of the problem

### 3.1    Multi-channel Convolutional Matrices

We focus our attention on *multichannel convolution matrices* that have *localized convolutional filters*. In this section, we introduce our notation and define the random matrix ensembles which are relevant to our result. We consider block structured signal vectors $x \in \mathbb{R}^{Pq}$ of the form $x = [x^{(i)}]_{i=1}^{P}$, and we refer to the blocks $x^{(i)} \in \mathbb{R}^q$ as 'channels.' For any vector of dimension $d$, we denote by $\mathcal{P}_d \in \mathbb{R}^{d \times d}$ the cyclic coordinate permutation matrix of order $d$, whose coordinates are $\langle e_i, \mathcal{P}_d e_j \rangle = \mathbf{1}[i = j+1]$. For a block-structured vector $x \in \mathbb{R}^{Pq}$, we denote by $\mathcal{P}_{P,q} \in \mathbb{R}^{Pq \times Pq}$ the block cyclic permutation matrix satisfying $(\mathcal{P}_{P,q} x)^{(i)} = x^{(i+1)}$ for $1 \leq i < P$, and $(\mathcal{P}_{P,q} x)^{(P)} = x^{(1)}$. Similarly, we denote by $\mathcal{S}_{i,j} \in \mathbb{R}^{Pq \times Pq}$ the swap permutation matrix which exchanges blocks $i, j$: $[\mathcal{S}_{i,j} x]^{(i)} = x^{(j)}$, $[\mathcal{S}_{i,j} x]^{(j)} = x^{(i)}$, and $[\mathcal{S}_{i,j} x]^{(k)} = x^{(k)}$ for $k \neq i, j$. Last, given a vector $\omega \in \mathbb{R}^k$ for $k \leq q$, denote by $\texttt{Zero-Pad}_{q,k}(\omega)$ the vector whose first $k$ coordinates are $\omega$, and whose other coordinates are zero.

$$\texttt{Zero-Pad}_{q,k}(\omega) = [\omega_1 \quad \omega_2 \quad \ldots \quad \omega_k \quad 0 \quad \ldots \quad 0] \in \mathbb{R}^q.$$

We define the following ensemble for random multi-channel convolution matrices.

**Definition 3.1** (Gaussian i.i.d. Convolution). Let $q \geq k$ be integers. The convolutional ensemble $\mathcal{C}(q, k)$ contains random circulant matrices $C \in \mathbb{R}^{q \times q}$ whose first row is given by $C_1 = \texttt{Zero-pad}_{q,k}[\omega]$ where $\omega \in \mathbb{R}^k$ has i.i.d. Gaussian coordinates $\omega_i \sim \mathcal{N}(0, 1/k)$. The remaining rows $C_i$ are determined by circulant structure, ie. $C_i = \mathcal{P}_q^{i-1} \texttt{Zero-pad}_{q,k}[\omega]$.

Random multi-channel convolutions are block-dense matrices with independent $\mathcal{C}(q, k)$ blocks.

**Definition 3.2** (Multi-channel Gaussian i.i.d. Convolution). Let $D, P \geq 1$ and $q \geq k \geq 1$ be integers. The random multi-channel convolution ensemble $\mathcal{M}(D, P, k, q)$ contains random block matrices $M \in \mathbb{R}^{Dq \times Pq}$ of the form

$$M = \frac{1}{\sqrt{P}} \begin{bmatrix} C_{1,1} & C_{1,2} & \ldots & C_{1,P} \\ C_{2,1} & \ddots & & \vdots \\ \vdots & & & \vdots \\ C_{D,1} & \ldots & & C_{D,P} \end{bmatrix}$$

where each $C_{i,j} \sim \mathcal{C}(q, k)$ is sampled independently.

Fig. 2 gives a graphical explanation of the link between these matrices and the convolutional layers. The parameter $P$ ($D$) is the number of input (output) channels, $q$ is the dimension of the input and $k$ the filter size.

| Layer | $D$ | $P$ | $q$ | $k$ |
|---|---|---|---|---|
| $1 \to 2$ | 1024 | 512 | $4^2$ | 1 |
| $2 \to 3$ | 512 | 256 | $8^2$ | 5 |
| $3 \to 4$ | 256 | 128 | $16^2$ | 5 |
| $4 \to 5$ | 128 | 3 | $64^2$ | 5 |

Figure 3: System sizes for convolutional layers in a DC-GAN architecture used to generate LSUN images [Radford et al., 2015, Figure 1]. These are *not* directly comparable to MCC matrices, as DCGAN uses *fractionally strided convolutions*, which can be thought of as a composition of an MCC matrix with superresolution. However, they give a reasonable picture of the sizes of typical layers in convolutional neural networks.

## 3.2 Thermodynamic-like Limit and Finite-size Regimes

We prove our main result in a thermodynamic-like limit $D, P \to \infty$ while $\beta = D/P$ is fixed and $q, k = O(1)$. From a practical perspective, convolutional layers in deep neural networks often use large channel dimensions ($D, P \gg 1$), large signal dimensions ($q \gg 1$), and a small filter size ($k = O(1)$). As an example, we show in Figure 3 the sizes of convolutional layers used by the DC-GAN architecture to generate LSUN images [Radford et al., 2015, Figure 1].

Interestingly, our theoretical predictions do not depend explicitly on the *relative* sizes of $q$ and $(D, P)$. We observe empirically that these predictions become accurate at finite sizes of $(D, P)$ which may seem small relative to $q$, and which are realistic from a practical neural network perspective. For example, in Figure 1, we observe strong empirical agreement with predictions for $q = P = 1024$ as $\beta$ and $D = \beta P$ vary.

## 3.3 Multi-layer AMP

In this section, we define a class of probabilistic graphical models (PGMs) that captures the inference problems of interest, and we state the Multi-layer Approximate Message Passing (ML-AMP) [Manoel et al. [2017]] iterations, which can be used for inference on these PGMs. We consider the following signal model.

**Definition 3.3** (Multi-layer Signal Model). Let $\{W^{(l)}\}_{1 \le l \le L}$ be matrices of dimension $W^{(l)} \in \mathbb{R}^{n_{l-1} \times n_l}$. Let $\{\varphi^{(l)}(z; \zeta)\}_{1 \le l \le L}$ be scalar channel functions for which $z$ is the estimation quantity and $\zeta$ represents channel noise. For vectors $z, \zeta \in \mathbb{R}^{n_{l-1}}$, we write $\varphi^{(l)}(z; \zeta)$ to indicate the coordinatewise application of $\varphi^{(l)}$. The multi-layer GLM signal model is given by

$$y = \varphi^{(1)}(W^{(1)}\varphi^{(2)}(W^{(2)}(\ldots \varphi^{(L)}(W^{(L)}x; \zeta^{(L)})\ldots).$$

We assume $x \in \mathbb{R}^{n_L}$ follows a known separable prior, $x_i \sim P_X(x)$ i.i.d., and that the channel noise $\zeta^{(l)}$ has i.i.d. $\mathcal{N}(0, (\sigma^{(l)})^2)$ coordinates of known variance $(\sigma^{(l)})^2 \ge 0$.

The full estimation quantities of the model are the coordinates of the vectors $\{h^{(l)}\}_{1 \le l \le L}$, $\{z^{(l)}\}_{1 \le l \le L}$, which are related by

$$y_\mu = \varphi^{(1)}(z^{(1)}; \zeta^{(1)}) \qquad\qquad z_\mu^{(1)} = \sum_i W_{\mu i}^{(1)} h_i^{(1)}, \qquad (2)$$

$$h_i^{(1)} = \varphi^{(2)}(z^{(2)}; \zeta^{(2)}) \qquad\qquad z_\mu^{(2)} = \sum_i W_{\mu i}^{(2)} h_i^{(2)},$$

$$\vdots$$

$$h_i^{(L-1)} = \varphi^{(L)}(z^{(L)}; \zeta^{(L)}) \qquad\qquad z_\mu^{(L)} = \sum_i W_{\mu i}^{(L)} x_i$$

where we take $h^{(L)} = x_0$. The corresponding conditional probabilities, which define the factor nodes of the underlying PGM, are given by

$$P^{(l)}(h \mid z) = \int d\zeta \, e^{-\frac{1}{2}\zeta^2} \delta(h - \varphi(z; \zeta)),$$

To compute the posterior marginals, ML-AMP iteratively updates the parameters of independent 1D Gaussian approximations to each marginal. Each coordinate $h_i^{(l)}(t)$ has corresponding parameters $\{A_i^{(l)}(t), B_i^{(l)}(t)\}$ and each $z_\mu^{(l)}(t)$ has corresponding $\{V_\mu^{(l)}(t), \omega_\mu^{(l)}(t)\}$, where $t \geq 1$ indexes the ML-AMP iterations. The recursive relationship between these parameters is defined in terms of scalar *denoising functions*, $\hat{h}^{(l)}$ and $g^{(l)}$, which compute posterior averages of the estimation quantities given their prior parameters.

In general, these denoising functions can be chosen (up to regularity assumptions) to adjust ML-AMP's performance in applied settings, such as in Metzler et al. [2015], and in these cases the denoisers may be nonseparable vector valued functions. However, in the separable, Bayes-optimal regime where $P_x(x)$ and $P^{(l)}(h \mid z)$ are known, the optimal denoisers are given by,

$$\hat{h}_i^{(l)}(t+1) := \partial_B \log \mathcal{Z}^{(l+1)}(A_i^{(l)}, B_i^{(l)}, V_i^{(l+1)}, \omega_i^{(l+1)}) \tag{3}$$

$$\sigma_i^{(l)}(t+1) := \partial_B \hat{h}_i^{(l)}(t+1)$$

$$g_\mu^{(l)}(t) := \partial_\omega \log \mathcal{Z}^{(l)}(A_\mu^{(l-1)}, B_\mu^{(l-1)}, V_\mu^{(l)}, \omega_\mu^{(l)})$$

$$\eta_\mu^{(l)}(t) := \partial_\omega g_\mu^{(l)}(t)$$

$$\mathcal{Z}^{(l)}(A, B, V, \omega) := \frac{1}{\sqrt{2\pi V}} \int P^{(l)}(h \mid z) \exp\left(Bh - \frac{1}{2}Ah^2 - \frac{(z-\omega)^2}{2V}\right) dh\, dz$$

where $2 \leq L \leq L-1$, $t \geq 2$ and the prior parameters on the right hand side are taken at iteration $t \geq 2$. The corresponding ML-AMP iterations are given by,

$$V_\mu^{(l)}(t) = \sum_i [W_{\mu i}^{(l)}]^2 \sigma_i^{(l)}(t) \qquad \omega_\mu^{(l)}(t) = \sum_i W_{\mu i}^{(l)} \hat{h}_i^{(l)}(t) - V_\mu^{(l)}(t) g_\mu^{(l)}(t-1) \tag{4}$$

$$A_i^{(l)}(t) = -\sum_\mu [W_{\mu i}^{(l)}]^2 \eta_\mu^{(l)}(t) \qquad B_i^{(l)}(t) = \sum_\mu W_{\mu i}^{(l)} g_\mu^{(l)}(t) + A_i^{(l)}(t)\hat{h}_i^{(l)}(t).$$

For the boundary cases $t = 1$, $l = 1$, and $l = L$, the iterations (3), (4) are modified as follows.

1. At $t = 1$, we initialize $B_i^{(l)} \sim P_{B_0}^{(l)}$ and $\omega_\mu^{(l)} \sim P_{\omega_0}^{(l)}$, where $P_{B_0}^{(l)}, P_{\omega_0}^{(l)}$ are the distributions of the signal model parameters (2) when $x_i \sim P_X$. We take $(A_i^{(l)})^{-1} = \text{Var}(B_i^{(l)})$ and $V_\mu^{(l)} = \text{Var}(\omega_\mu^{(l)})$.

2. At $l = 1$, the denoiser $g_\mu^{(1)}(t) = \partial_\omega \log \mathcal{Z}^{(1)}(y, V_\mu^{(1)}, \omega_\mu^{(1)})$, where

$$\mathcal{Z}^{(1)}(y, V_\mu^{(1)}, \omega_\mu^{(1)}) = \frac{1}{\sqrt{2\pi V}} \int P^{(1)}(y \mid z) \exp\left(-\frac{(z - \omega_\mu^{(1)})^2}{2V_\mu^{(1)}}\right) dz.$$

3. At $l = L$, the denoiser $\hat{h}^{(L)}(t) = \partial_B \log \mathcal{Z}^{(L)}(A_i^{(L)}, B_i^{(L)})$, where

$$\mathcal{Z}^{(L)}(A_i^{(L)}, B_i^{(L)}) = \int P_X(h) \exp\left(B_\mu^{(L)} h - \frac{1}{2} A_\mu^{(L)} h^2\right) dh.$$

### 3.3.1 Computational Savings of MCC Matrices

As ML-AMP requires only matrix-vector products, its computational burden can be significantly reduced when using structured and/or sparse sensing matrices. In our setting, multi-channel convolutions $M \sim \text{MCC}(D, P, q, k)$ have $DPk$ nonzero coordinates, compared to $DPq^2$ nonzero coordinates of a Gaussian i.i.d. matrix. Typically, $k$ represents the size of a localized filter applied to a larger image, with $k \ll q$ [Gonzalez and Woods, 2008, Section 3.4], leading to significant space savings by a factor $k/q^2$. This same is true in convolutional neural networks, where the use of localized convolutional filters represents an inductive bias towards localized features that is considered a key aspect of their practical success Krizhevsky et al. [2012], Zeiler and Fergus [2014].

In addition to space savings, specialized matrix-vector product implementations can reduce the time complexity of ML-AMP with MCC sensing matrices. Simple routines for sparse matrix-vector products run in time proportional to the number of nonzero entries, resulting in the same $k/q^2$ constant factor speed up for MCC matrix-vector products. Alternatively, if $k \gg \log q$, then a simple algorithm using a fast Fourier transform for convolution-vector products yields time complexity $O(DPq \log q)$. Such an algorithm is sketched Appendix B.

## 4 Main result

We now state our main technical result, starting with the set of required assumptions.

(A1) for any $1 \leqslant l \leqslant L$, the function $\varphi^{(l)}(z; \zeta)$ is continuous and there exists a polynomial $b^{(l)}$ of finite order such that, for any $x \in \mathbb{R}$, $|\varphi^{(l)}(x; \zeta)| \leqslant |b^{(l)}(x, \zeta)|$.

(A2) for any $1 \leqslant l \leqslant L$, the matrix $\mathbf{W}^{(l)}$ is sampled from the ensemble $\mathcal{M}(D^l, P^l, k^l, q^l)$ where $P^l q^l = D^{l-1} q^{l-1}$

(A3) the iteration (4) is initialized with a random vector independent of the mixing matrices verifying $\frac{1}{N} \|\mathbf{h}_0\|_2^2 < +\infty$ almost surely, and the prior distribution $P_X$ is subGaussian

(A4) for any $1 \leqslant l \leqslant L$, $D_l, P_l \to \infty$ with constant ratio $\beta_l = D_l/P_l$, with finite $q_l$.

Under these assumptions, we may define the following *state evolution* recursion

**Definition 4.1** (State Evolution). Consider the following recursion,

$$\hat{\kappa}^{(l)}(t) = -\beta^{(l)} \mathbb{E}^{(l)}[\partial_\omega g(\hat{\kappa}^{(l-1)}, b, \tau_1 - \kappa^{(l)}, h)] \tag{5}$$

$$\kappa^{(l-1)}(t+1) = \mathbb{E}^{(l)}[h \, \hat{h}^{(l-1)}(\hat{\kappa}^{(l-1)}, b, \tau_1 - \kappa^{(l)}, h)], \tag{6}$$

where $\tau^{(l)}$ is the second moment of $P_{B_0}^{(l)}$, where the right hand side parameters are taken at time $t$, and the expectations $\mathbb{E}^{(l)}$ are taken with respect to

$$P^{(l)}(w, z, h, b) = P_{\text{out}}^{(l)}(h \mid z) \mathcal{N}(z; w, \tau^{(l)} - \kappa^{(l)}) \mathcal{N}(w; 0, \kappa^{(l)}) \mathcal{N}(b; \hat{\kappa}^{(l-1)} h, \hat{\kappa}^{(l-1)}).$$

At $t = 1$, the state evolution is initialized at $\kappa^{(l)} = 0$ and $(\hat{\kappa}^{(l)})^{-1} = \tau^{(l)}$. At the boundaries $l = 1, L$, the expectations are modified analogously to the ML-AMP iterations as described by Manoel et al. [2017]. We then have the following asymptotic characterization of the iterates from the convolutional ML-AMP algorithm

**Theorem 4.2.** Under the set of assumptions (A1)-(A4), for any sequences of uniformly pseudo-Lipschitz functions $\psi_1^N, \psi_2^N$ of order $k$, for any $1 \leqslant l \leqslant L$ and any $t \in \mathbb{N}$, the following holds

$$\frac{1}{D_l q_l} \sum_{i=1}^{D_l q_l} \psi_1(\omega_i^{(l)}(t), B_i^{(l-1)}(t)) \xrightarrow{\text{P}} \mathbb{E}\left[\psi_1\left(Z^{(l)}(t), \hat{Z}^{(l-1)}(t)\right)\right] \tag{7}$$

$$\frac{1}{P_l q_l} \sum_{i=1}^{P_l q_l} \psi_2(\omega_i^{(l+1)}(t), B_i^{(l)}(t)) \xrightarrow{\text{P}} \mathbb{E}\left[\psi_2\left(Z^{(l+1)}(t), \hat{Z}^{(l)}(t)\right)\right] \tag{8}$$

where $Z^l(t) \sim \mathcal{N}(0, \kappa^l(t))$, $\hat{Z}^l(t) \sim \mathcal{N}(0, \hat{\kappa}^l(t))$ are independent random variables.

### 4.1 Proof Sketch

The proof of Theorem 4.2, which is given in Appendix A, has two key steps. First, we construct permutation matrices $U, \tilde{U}$ such that for $W \sim \text{MCC}(D, P, q, k)$, the matrix $\tilde{W} = UW\tilde{U}^T$ is a block matrix whose blocks either have i.i.d. Gaussian elements or are zero valued, and has a block-circulant structure. The effect of the permutation is that entries of $\tilde{W}$ which are correlated due to circulant structure of $W$ are relocated to different blocks. Once these permutation matrices are defined, we define a new, matrix-valued AMP iteration involving the dense Gaussian matrices obtained from the permutations, and whose non-linearities account for the block-circulant structures and the permutation matrices. The state evolution of this new iteration is proven using the results of Gerbelot and Berthier [2021]. This provides an explicit example of how the aforementioned results can be used to obtain rigorous, non Bayes-optimal SE equations on a composite AMP iteration. The separability assumption is key in showing that the AMP iterates obtained with the convolutional matrices can be *exactly* embedded in a larger one. Note that this is a stronger result than proving SE equations for an algorithm that computes marginals of a random convolutional posterior: we show the SE equations are the same as in the dense case. We finally invoke the Nishimori conditions, see e.g. Krzakala et al. [2012], to simplify the generic, non Bayes-optimal SE equations to the Bayes-optimal ones.

$$
\left[\begin{array}{ccc|ccc|ccc}
z_{11} & w_{11} & & z_{12} & w_{12} & & z_{13} & w_{13} & \\
 & z_{11} & w_{11} & & z_{12} & w_{12} & & z_{13} & w_{13} \\
w_{11} & & z_{11} & w_{12} & & z_{12} & w_{13} & & z_{13} \\
z_{21} & w_{21} & & z_{22} & w_{22} & & z_{23} & w_{23} & \\
 & z_{21} & w_{21} & & z_{22} & w_{22} & & z_{23} & w_{23} \\
w_{21} & & z_{21} & w_{22} & & z_{22} & w_{23} & & z_{23} \\
z_{31} & w_{31} & & z_{32} & w_{32} & & z_{33} & w_{33} & \\
 & z_{31} & w_{31} & & z_{32} & w_{32} & & z_{33} & w_{33} \\
w_{31} & & z_{31} & w_{32} & & z_{32} & w_{33} & & z_{33} \\
z_{41} & w_{41} & & z_{42} & w_{42} & & z_{43} & w_{43} & \\
 & z_{41} & w_{41} & & z_{42} & w_{42} & & z_{43} & w_{43} \\
w_{41} & & z_{41} & w_{42} & & z_{42} & w_{43} & & z_{43}
\end{array}\right]
\qquad
\left[\begin{array}{ccc|ccc|ccc}
z_{11} & z_{12} & z_{13} & w_{11} & w_{12} & w_{13} & & & \\
z_{21} & z_{22} & z_{23} & w_{21} & w_{22} & w_{23} & & & \\
z_{31} & z_{32} & z_{33} & w_{31} & w_{32} & w_{33} & & & \\
z_{41} & z_{42} & z_{43} & w_{41} & w_{42} & w_{43} & & & \\
 & & & z_{11} & z_{12} & z_{13} & w_{11} & w_{12} & w_{13} \\
 & & & z_{21} & z_{22} & z_{23} & w_{21} & w_{22} & w_{23} \\
 & & & z_{31} & z_{32} & z_{33} & w_{31} & w_{32} & w_{33} \\
 & & & z_{41} & z_{42} & z_{43} & w_{41} & w_{42} & w_{43} \\
w_{11} & w_{12} & w_{13} & & & & z_{11} & z_{12} & z_{13} \\
w_{21} & w_{22} & w_{23} & & & & z_{21} & z_{22} & z_{23} \\
w_{31} & w_{32} & w_{33} & & & & z_{31} & z_{32} & z_{33} \\
w_{41} & w_{42} & w_{43} & & & & z_{41} & z_{42} & z_{43}
\end{array}\right]
$$

Figure 4: A sketch of the permutation lemma applied to matrix $W \sim \mathrm{MCC}(4,3,3,2)$. Left: $W$ before permutation. Right: after permutation, $UW\tilde{U}^T$.

The idea of embedding a non-separable effect such as a block-circulant structure or different variances in a mixing matrix is the core idea in the proofs of SE equations for spatially coupled systems, notably as done in Javanmard and Montanari [2013], Donoho et al. [2013]. We note that in the numerical experiments shown at Figure 1, the parameter $q$, considered finite in the proof, is actually comparable to the number of channel, considered to be extensive. Empirically we observe that this does not hinder the validity of the result, something that was also observed in the spatial coupling literature, e.g. Krzakala et al. [2012], where large number of different blocks in spatially coupled matrices were considered, with convincing numerical agreement.

The existence of permutations matrices verifying the property described above is formalized in the following lemma:

**Lemma 4.3** (Permutation Lemma). Let $W \sim \mathcal{M}(D, P, k, q)$ be a multi-channel convolution matrix. There exist row and column permutation matrices $U \in \mathbb{R}^{Dq \times Dq}$, $\tilde{U} \in \mathbb{R}^{Pq \times Pq}$ such that $\tilde{W} = UW\tilde{U}^T$ is a block-convolutional matrix with dense, Gaussian i.i.d. blocks. That is,

$$
\tilde{W} = \frac{1}{\sqrt{k}}
\begin{bmatrix}
A^{(1)} & A^{(2)} & \dots & A^{(k)} & & & \\
 & A^{(1)} & A^{(2)} & \dots & A^{(k)} & & \\
\vdots & & A^{(2)} & \dots & A^{(k)} & & \\
 & & & & \ddots & & \vdots \\
A^{(2)} & A^{(3)} & \dots & A^{(k)} & & & A^{(1)}
\end{bmatrix}
$$

where each $A^{(s)} \in \mathbb{R}^{D,P}$, $1 \le s \le k$ has i.i.d. $\mathcal{N}(0, 1/P)$ coordinates.

*Proof.* Consider the elements of the matrix $M$ which are non-zero and sampled i.i.d. as opposed to exact copies of other variables. They are positioned on the first row of each block of size $q \times q$, and thus the indexing for their rows and columns can be written as $M_{aq+1, bq+c}$ where $a, b, c$ are integers such that $0 \le a \le D-1$, $0 \le b \le P-1$ and $1 \le c \le k$. The integers $a, b$ describe the position of the $q \times q$ block the variable is in, and $c$ describes, for each block, the position in the initial random Gaussian vector of size $k$ that is zero-padded and circulated to generate the block. The goal is to find the mapping that groups these variables into $k$ dense blocks of extensive size $D \times P$. To do so, one can use the following bijection $\tilde{M}_{\gamma, \alpha P + \beta} = M_{aq+1, bq+c}$ where $\gamma = a + 1$, $\alpha = c - 1$ and $\beta = b + 1$. By doing this, $c$ becomes the block index and $a, b$ become the position in the dense block. This mapping can be represented by left and right permutation matrices which also prescribe the permutation for the rest of the elements of $M$. Figure 4 shows a sketch of this permutation. $\square$

The state evolution recursion given in Definition 4.1 is equivalent to the state evolution for unstructured matrices proposed by Manoel et al. [2017], which may seem surprising relative to existing literature on spatial coupling, where spatially coupled sensing matrices are used to achieve improved performance over unstructured sensing matrices. This equivalence is a consequence of the fact that $\tilde{W}$ has dense Gaussian blocks which each have the same variance, whereas in the literature, spatially coupled matrices typically have Gaussian blocks with inhomogeneous variances that can be tuned to improve recovery performance.

## 5 Numerical Experiments

In this section, we compare state evolution predictions from Theorem 4.2 with a numerical implementation of the ML-AMP algorithm described in Section 3.3. Our first experiment, shown in Figure 1, is a noisy compressive sensing task under a sparsity prior $P_X(x) = \rho \mathcal{N}(x; 0, 1) + (1 - \rho)\delta(x)$, where $\rho$ is the expected fraction of nonzero components of $x_0$. Measurements are generated $y_0 = Wx_0 + \eta$ for noise $\eta \sim \mathcal{N}(0, 10^{-4})$, where $W \sim \text{MCC}(D, P, q, k)$. We show recovery performance at sparsity levels $\rho \in \{0.25, 0.5, 0.75\}$ as the measurement ratio $\beta = D/P$ varies, averaged over 10 independent AMP iterates. Additionally, we show convergence of the (averaged) AMP iterates for sparsity $\rho = 0.25$ at a range of $\beta$ near the recovery threshold. We observe strong agreement between AMP empirical performance and the state evolution prediction. The system sizes are $P = 1024$, $q = 1024$, with $D = \beta P$ varying.

In Figure 5, we consider two examples of $L = 2, 3, 4$ layer models following Equation (2). In both, the output channel $l = 1$ generates noisy, compressive linear measurements $y = z^{(1)} + \zeta$ for $\zeta_i \sim \mathcal{N}(0, \sigma^2)$ and for dense couplings $W_{ij}^{(1)} \sim \mathcal{N}(0, 1/n^{(1)})$. Layers $2 \le l \le 4$ use MCC couplings $W^{(l)} \sim \text{MCC}(D_l, P_l, q, k)$, where $qP_l = n_l$ and $D_l = \beta P_l = qn_{l-1}$. Channel functions $\{\varphi^{(l)}\}$ vary across the two experiments. The input prior is $P_X(x) = \mathcal{N}(x; 0, 1)$ and the problem parameters are $q = 10$ channels, filter size $k = 3$, noise level $\sigma^2 = 10^{-4}$, input dimension $n^{(L)} = 5000$, and layerwise aspect ratios $\beta^{(L)} = 2$ and $\beta^{(l)} = 1$ for $2 \le l < L$. Finally, the channel aspect ratio $\beta^{(1)}$ varies in each experiment.

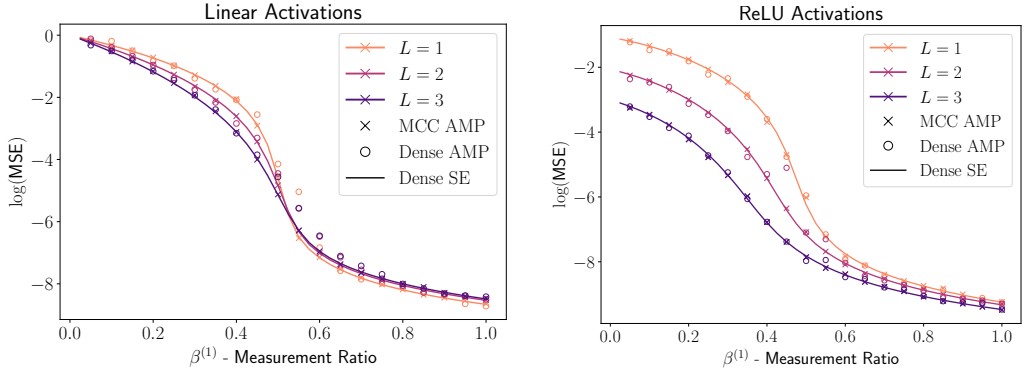

Figure 5: ML-AMP compressive sensing recovery under multichannel convolutional designs (crossed) and the state evolution for the corresponding fully connected model (lined). For comparison, we also plot the corresponding fully connected AMP iterations (dotted), in which $W^{(l)} \in \mathbb{R}^{D_l \times P_l}$ with $W_{ij} \sim \mathcal{N}(0, 1/P_l)$, with the dimensions of the prior and output channel adjusted appropriately. Left: For $2 \le l \le L$, the channel functions are $\varphi^{(l)}(z; \zeta) = z + \zeta$ where $\zeta_i \sim \mathcal{N}(0, \sigma^2)$. Right: For $2 \le l \le L$, the channel functions are $\varphi^{(l)}(z; \zeta) = \max(z, 0)$ where the maximum is applied coordinatewise. This channel function is the popular ReLU activation function used by generative convolutional neural networks such as in Radford et al. [2015], Bora et al. [2017].

We compare the state evolution equations to empirical AMP results in two cases. In the left panel, we show multilayer models with identity channel functions, and in the right panel, we show models with ReLU channel functions. The latter captures a simple but accurate example of a convolutional GNN.

## 6 Discussion and Future Work

We have proven state evolution recursions for the ML-AMP algorithm for signal recovery from multilayer convolutional networks. We consider networks whose weight matrices are drawn either i.i.d. Gaussian or from an ensemble of random multi-channel convolution matrices. Interestingly, under a separable prior and separable channel functions, these two matrix ensembles yield the same state evolution (up to a rescaling). These predictions closely match empirical observations in compressive sensing under a sparsity prior (Figure 1) and under multi-layer priors (Figure 5).

Based on the discussion in Section 4.1, the equivalence property is expected to break down if $\tilde{W}$ is replaced by a block matrix $\tilde{W}_{\text{sp}}$ with dense Gaussian blocks organized in the same block-circulant structure, with inhomogeneous variances. As we discuss in Appendix D, by inverting the permutation lemma, $\tilde{W}_{\text{sp}}$ corresponds to a multi-channel convolution $W_{\text{sp}}$ with inhomogeneous convolutional filters, which can be taken as a simple model for structured convolutions. Inhomogeneous blocks can also easily be incorporated in the Graph-AMP framework using nonseparable denoisers, so the key elements of our proof also apply to the inhomogeneous case. This generalized model for MCC matrices represents an interesting direction for future exploration, whose state evolution is expected to diverge from that of the dense Gaussian ensemble.

## Acknowledgments and Disclosure of Funding

M.D. acknowledges funding from Northeastern University's Undergraduate Research and Fellowships office and the Goldwater Award. We also acknowledge funding from the ERC under the European Union's Horizon 2020 Research and Innovation Program Grant Agreement 714608- SMiLe, and by the Swiss National Science Foundation grant SNFS OperaGOST, 200021_200390.

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
