# OpenReview forum: "Multi-layer State Evolution Under Random Convolutional Design"
_NeurIPS.cc/2022/Conference — NeurIPS 2022 Accept_

### Official Review · Reviewer_paFy · 2022-07-08

**Rating:** 7
**Confidence:** 4
**Soundness:** 3 good
**Presentation:** 3 good
**Contribution:** 3 good

**Summary:**

The submitted paper addresses multi-layer aproximate message-passing (ML-AMP) for signal recovery under multi-layer generative neural network priors (1), in which the activation (or denoiser in compressed sensing) is separable and the sensing matrices are sampled from the multi-channel convolution ensemble in Definition 3.2. ML-AMP was originally proposed by Manoel et al. [2017] for dense sensing matrices with zero-mean i.i.d. (Gaussian) elements. The submitted paper generalizes the existing ML-AMP to the convolutional case in Definition 3.2.

The main contribution is rigorous state evolution of ML-AMP for the multi-channel convolution ensemble, i.e. Theorem 4.2. The proof is fully based on graph-based AMP and its rigorous state evolution by Gerbelot and Berthier [2021]. The main technical contribution is Lemma 4.3, allowing us to transform any multi-channel convolution matrix into block-convolutional matrix with dense, zero-mean, and i.i.d. Gaussian blocks via permutation. When separable activations, i.i.d. signals, and i.i.d. channel noise in (2) are assumed, all  permutation matrices are absorbed into messages in graph-based AMP via the change of variables. Since graph-based AMP is general enough to model ML-AMP, ML-AMP for the multi-channel convolution ensemble reduces to a special case of graph-based AMP.

The state evolution result is also justified via numerical simulations for artificial data. No numerical simulations are presented to show the advantage of the multi-channel convolution ensemble against conventional dense Gaussian ensemble or practical usefulness of ML-AMP for real data.

**Questions:**

--(A1) assumes pseudo-Lipschitz activations of any finite order while conventional state evolution by Bayati and Montanari [2011] postulated the Lipschitz-continuity. When pseudo-Lopschitz activations are used, in my understanding, the boundedness of higher-order signal moments is required as the iteration t increases. Eventually, we need to assume the boundedness of all signal moments. Do not the authors need to assume any regularity conditions on the signal prior, such as sub-Gaussian?

--The terminology "line" is a bit unclear. Clarify whether it means column or row.

--\tilde{W} in Lemma 4.3 is not displayed appropriately. It should be presented like (57) in Appendix A.

**Limitations:**

The submitted paper is theoretical and therefore has no negative social impacts.

**Strengths And Weaknesses:**

Strength
The strength is the proof of Theorem 4.2. From a technical point of view, the main part of the contributions to prove the theorem might be in Gerbelot and Berthier [2021]. Nonetheless, it should be significant contributions to understand graph-based AMP and apply its framework to signal recovery under multi-layer generative neural network priors. If the framework were old results, such as Bayati and Montanari [2011] or Rangan et al. [2019], I would not have positive impression since they are (well) known in the NeurIPS community. However, graph-based AMP is state-of-the-art message-passing and therefore not fully understood in the community.

Weakness
As pointed out in Summary, the weakness of the submitted paper presents no numerical simulations to show the advantage of the multi-channel convolution ensemble against conventional dense Gaussian ensemble or practical usefulness of ML-AMP for real data. The paper only presents numerical simulations for artificial data with the so-called Bernoulli-Gaussian prior. At least the authors could add numerical comparisons between the multi-channel convolution ensemble and conventional dense Gaussian ensemble for artificial data.

---

> ### Author Response · Authors · 2022-08-01
> **Response to R1-PaFY**
>
> Thank you for your comments on our work. In response to the questions and weaknesses identified for this paper:
>
> - __Lack of numerical simulations__: we will add empirical comparisons between the Gaussian ensemble and the MCC ensemble to both Figure 1, for the Gauss-Bernoulli prior, and Figure 5, where we study artificial data generated by multilayer linear and ReLU nets. Specifically, for each ‘x’ (MCC AMP empirical performance), we will add an ‘o’ indicating the performance of the corresponding model with dense Gaussian weights. Since these two models admit the same state evolution equations in the separable case, their performance is expected to be nearly identical (see eg. Manoel et al. [2017] Figure 2, agreement of dense Gaussian AMP and SE).
>
> - __Benefits of MCC over dense Gaussian__: The main advantage of the convolutional ensemble is its sparsity, which makes the resulting convolutional ML-AMP significantly faster. As discussed in Section 3.3.1, MCC matrix multiplications have a speedup by factor $O(k/q^2)$ over the Gaussian ensemble, which is a significant practical improvement (eg. in Figure 5, $q=10$, $k=3$, so roughly $30x$ speedup per matrix multiplication). We will highlight this more clearly in the body and introduction of the paper.
>
>   Relative to previous work on ML-AMP, which require strong independence assumptions on the sensing matrices, our contribution is to prove SE for a model with sparse, structured sensing matrices. Because of this structure, the corresponding AMP algorithm is significantly faster while having the same state evolution, which is one practical advantage of the MCC ensemble over the Gaussian ensemble.
>
> - __Pseudo-Lipschitz regularity of the activations__: the framework of Gerbelot-Berthier 2021 shows that the activations can be pseudo-lipschitz of any order since the state evolution variables are Gaussian, which can be integrated against any polynomial. However, because the proof of Gerbelot-Berthier 2021 is written in a non-separable framework, the definition of pseudo-Lipschitz function is scaled differently than the original one from Bayati-Montanari, which mainly changes the definition of low-dimensional observables, see Definition A1 and the comment below. We indeed need to assume that the prior distribution $p_x$ is subGaussian in order for higher moments to be well defined. We thank the reviewer for spotting this inconsistency and will correct it in the revised version.
>
>
> - __Clarity & typos__: We will fix the typo in Lemma 4.3. By ‘line,’ we mean ‘row,’ and we will update the text to use only the latter.

---

> > ### Comment · Reviewer_paFy · 2022-08-08
> > **Response to authors' rebuttle**
> >
> > Thank you for your responses. I do not intend to evaluate your paper negatively. I agree with the other reviewers who recommended 6: weak accpet or 7: accept.

---

> > > ### Comment · Reviewer_paFy · 2022-08-10
> > > **My score**
> > >
> > > I have increased my score in terms of the fairness between the other papers I reviewed.

---

### Official Review · Reviewer_fcUP · 2022-07-11

**Rating:** 7
**Confidence:** 1
**Soundness:** 4 excellent
**Presentation:** 4 excellent
**Contribution:** 3 good

**Summary:**

This paper provides a theoretical study on the multi-layer approximate message passing (ML-AMP) algorithm with sensing matrices in some of the layers being multi-channel convolution, which extends previous results with regular 2D sensing matrices in all layers. The main result is the state evolution equations presented in Sec. 4, derived under asymptotic limits on the shape of convolution matrices. Experimental evidence shows that the theoretical results align well with the simulation results.

**Questions:**

The paper may benefit from a discussion on why their theoretical result is of practical interests.

**Limitations:**

This is discussed at the end of the paper.

**Strengths And Weaknesses:**

**Originality**

As the paper explains, there is existing work on the analysis of ML-AMP for regular (2D) sensing matrices. The novelty of this work lies in extending such analysis for the case where the sensing matrices are convolutional.

**Quality**

The paper seems to be providing a good quality theoretical result that aligns well with the simulation results, even though the analysis concerns an asymptotic case only.

**Clarity**

I find the paper to be very well written, with an informative introduction, sufficient background before presenting their main result and so on.

---

> ### Author Response · Authors · 2022-08-01
> **Response to R2-fcUP**
>
> Thank you for reviewing our work. As discussed with **R1-paFy**, part of the practical interest of our theoretical result is that MCC matrices are computationally cheaper to use in ML-AMP than their Gaussian counterparts because of their sparse circulant structure. We plan to highlight this more clearly in our experiments and introduction. MCC matrices are also practically interesting because of their popularity in areas like deep learning, computer vision, and image recovery. It was not a-priori clear that these structured matrices are compatible with AMP, since existing state evolution proofs require strong randomness assumptions on the sensing matrices.

---

### Official Review · Reviewer_LN3c · 2022-07-18

**Rating:** 6
**Confidence:** 4
**Soundness:** 3 good
**Presentation:** 2 fair
**Contribution:** 3 good

**Summary:**

This paper studies the problem of estimating a signal vector from an observation obtained via a multilayer convolutional model. The weights defining the convolutional model are assumed to be random (and known). The main contributions are an AMP algorithm for signal estimation and a theorem proving that state evolution accurately predicts the performance of AMP on this model.  Numerical results, presented for synthetic settings, show good agreement of  empirical performance and state evolution predictions.

**Questions:**

-- It is mentioned in the abstract and the introduction that the state evolution equations for the MCC model are the same for as the iid Gaussian case, upto rescaling. I find this surprising based on an analogy with spatial coupling, where AMP with spatially coupled matrices has a different state evolution from the iid Gaussian case. The authors should discuss this point after Definition 4.1/Theorem 4.2, and if possible, give some high-intuition on why the two SEs are the same.

-- The blue shaded parts in Fig. 2 are confusing.  It's not clear why kP entries from the matrix on the left are connected to a single one on the right.

-- the notation is Definition 4.1 (State evolution) is inconsistent with the theorem below. One uses m and \hat{m}, while the other uses \kappa and \hat{\kappa}. Do these refer to the same quantities?

-- To have a state evolution result that can applied to compute asymptotic loss, like MSE, the test functions in Theorem 4.2 should also take as arguments h^l/x or z^l. Please also refer to the assumptions (on x and the noise) in Def. 3.3 in the theorem statement.

-- In the definition of the multilayer model (in the display above eq. 2), the same noise variable  \eta is used for each layer. If the \eta's in each layer are indepdendent, please make this clear.

-- The noise level \eta in the simulations is chosen to be 10^{-4}, extremely small. Is this required to get a good match between empirical results and SE at reasonable dimensions?

-- Below eq. (2), could you clarify whether h^L is taken to be x?

Typos:
l.33 : distributions --> distribution;   l.206: should be "iteration (4)" -- add parentheses around 4.

**Limitations:**

The random convolutional model is simplistic and omits many aspects of real convolutional networks, but the authors acknowledge this.

**Strengths And Weaknesses:**

The technical contribution of extending the multilayer AMP and state evolution from iid Gaussian weight matrices to a specific class of random convolutional matrices is a nice one. The proof technique is clever: it uses permutatation matrices to shuffle the weight matrices so that the correlated entries are in separate blocks, and then invokes the graph-based AMP proposed to define a matrix-valued AMP for the permuted matrix.  The problem setting is a bit artificial since assumes that the weights are random (rather than learned from data), but in my opinion the novelty of the proof technique is a nice contribution.

A weakness of the paper is the presentation. The paper is dense, some of which is due to the notation required to defined the convolutional matrices, but the notation is inconsitent and sometimes confusing. Key results and their implications should also be discussed in more depth to aid understanding. More details in the "Questions" section below.

---

> ### Author Response · Authors · 2022-08-01
> **Response to R3-LN3C**
>
> Thank you for your comments on our work. We agree that the paper will be clearer with additional discussion of the key results and implications, we will add this to the Introduction and Main Results. To answer the questions:
>
> - __Equivalence of Gaussian and MCC state evolution__: we agree that our equivalence result is somewhat surprising in the context of existing literature on spatial coupling, where spatially coupled sensing matrices are used to achieve better recovery thresholds than their dense counterparts. The reason that MCC state evolution reduces to the dense case is because, according Lemma 4.3, the matrix $\tilde{W}$ has dense Gaussian blocks which each have the _same_ variance. This is different from the literature on spatial coupling, where different Gaussian blocks have different variances and correspondingly a different SE.
>
>     So, in order to break the equivalence, one would need to use $\tilde{W}$ with Gaussian blocks of inhomogenous variances. This can actually be represented in the Graph AMP framework, and our proof would go through easily, but for simplicity we focus on the homogeneous case. Specifically, while we focus on separable denoisers (scalar functions applied coordinatewise to vectors), one can represent inhomogeneous variances using a non-separable denoising function that applies a different scaling to different coordinates of its input vector. We comment on this idea in Appendix D, where we also show how inhomogeneous variances can be used to represent a simple model for structured convolutions.
>
>     We will add further discussion following Theorem 4.1 to highlight the above intuition and the contrast between our model and the spatial coupling literature. We will also add further discussion of the ideas in Appendix D to the body of our work.
>
> - __Figure 2 Caption__: for $W \sim \mathcal{M}(D, P, q, k)$, each row of $W$ has $kP$ nonzero entries, because each row of $W$ is a concatenation of $P$ different rows from convolutional matrices, each row having $k$ nonzero entries. This figure is intended to show diagrammatically which $kP$ coordinates of an input signal $x$ contribute to a single coordinate of an output signal $y = Wx$. We will clarify this point in the description.
>
> - __Definition 4.1 and Theorem 4.2__:  yes, $(m, \hat{m})$ and $(\kappa, \hat{\kappa})$ refer to the same quantities. We have fixed this and the other highlighted typos in these two statements.
>
> - __Definition of the multilayer model__: yes, the $\zeta$ are independent between different layers, we will clarify this in the definition of the multilayer model.
>
> - __Noise level__: no, $\sigma^2 = 10^{-4}$ is not required for empirical agreement at reasonable sizes. We will note that. We keep this value fixed in our experiments for no particular reason. There are many examples in the ML-AMP literature of agreement between empirical statistics and SE predictions for a variety of channel noise levels, such as:
>    - Manoel et al. [2017], Figure 2, studies sparse linear regression and perceptron models at noise $\sigma^2= 10^{-8}$.
>    - Rangan [2010], figure 4, studies sparse generalized linear regression at noise $\sigma^2 = 10^{-2}$. (Available at [https://arxiv.org/abs/1010.5141](https://arxiv.org/abs/1010.5141))

---

> > ### Comment · Reviewer_LN3c · 2022-08-09
> > **Reply to authors**
> >
> > Thanks very much for the replies. The equivalence of the Gaussian and MCC state evolution due to the variance being the same in each block makes sense. Thanks again for the clarification.

---

### Official Review · Reviewer_xNsM · 2022-07-25

**Rating:** 7
**Confidence:** 4
**Soundness:** 3 good
**Presentation:** 3 good
**Contribution:** 3 good

**Summary:**

The paper establishes state evolution analysis of multi-layer approximate message passing algorithms used to find an input vector from multi-layer noisy measurements (or finding the latent code of generative models) with random convolutional layers or random Gaussian layers.  The proof is based on writing random convolutional layers in terms of random Gaussian matrices using permutation matrices (Lemma 4.3). The proof points to a connection between convolutional matrices and spatially coupled matrices in coding theory. The results are numerically verified for linear and multi-layer cases.


**Questions:**

* Regarding Definition 3.3, Generative priors do not have noise in them. Note that this is not a bias term, otherwise, the conditional distributions  like $P^{(l)}(h|z)$ are not needed. This can indicate that the paper is about multi-layer measurements and not generative priors. The authors should comment on what happens if the channel noise is assumed to be zero or non-random. If I am not mistaken, the analysis should remain valid (at least in $y=Ax$ case, where the Onsager correction disappears).


* The problem description of the paper is bit confusing specially when the connection with generative prior is mentioned. Generative priors  are used to parametrize a signal $s$ using $s=G_\theta(x)$, and then solve an inverse problem $y=As$ using it, for example by gradient descent on $\Vert y-AG_\theta(x) \Vert_2^2$.

The authors. However, mention: “one seeks to recover a data signal $x_0$ given access to measurements $y_0 = G_\theta(x_0)$”. This seems to combine the generative prior and measurement model.  The measurements are typically of form $y=Ax$; one can try to incorporate it as the last layer of the generative model, which is implicitly done by the authors. See Page 1, and the choice $\phi(z)=z$ which means $y=Wh$. However, this is not typical generative compressed sensing setup.  As I reflected in my summary, the paper can be seen as finding the latent code of a generative model, which can be useful in context of generative compressed sensing. In any case, there is a subtle difference between these cases, and the authors can comment to clarify it.

* As far as I can see Manoel et al. 2017 consider a general random matrix with i.i.d. entries and not only Gaussian (in contrast with the claim in page 2). However, the matrices are still unstructured. It is good if the authors replace the term “dense Gaussian” with a more general term, say unstructured random matrices.

* The last paragraph of “Related Work” section: the notion of stable training, mentioned in this paragraph, is a bit convoluted and ill defined. The stability issues of training GANs is very different from covariate shift related stability handled by batch norm. These are different notions of stability. Besides, I do not see the relevance of this discussion for the rest of the paper.

* On flow based models, it is better to cite the original work of Rezende et al 2015 and Dinh et al. 2015 and then mention follow up works like GLOW, RealNVP.

* Note that Bora et al do not only consider convolutional models. They also use MLP variants of generative priors (VAEs).

**Limitations:**

The main limitation of the paper is the way it connects with existing works on generative priors.

**Strengths And Weaknesses:**

**Strengths**

* Applying state evolution analysis in deep learning context is very interesting direction, which is explored here. Similarly, spatially coupled matrices and related theoretical works are an important and rich area of coding theory, and the connection with coding theory opens up an interesting perspective.
* The paper is in general well written and covers related works in a clear and adequate way.
* The proofs and derivations seem sound to me and tend to follow standard message passing derivations apart from Lemma 4.3. I have not checked the derivations line by line.

**Weaknesses**

* The problem considered in this paper and some of the assumptions are different from typical generative compressed sensing paper (putting aside the assumptions on randomness of parameters). Please see my comments below.
* The core issue, related to the comment above, is to mix the notion of measurements and the notion of prior. Generative priors are not measurements in context of generative CS. In the paper, the layers of the generative model are measurements with their measurement noise (channel noise $\zeta$ in Definition 3.3). See my comments below on the notion of channel noise.
* Practical models do not have random convolutions but trained ones. The implication of the current approach for the trained models is not clear and not discussed in the paper.

Overall, the paper has a valuable contribution and present that contribution properly, however, the connection with generative compressed sensing in style of Bora et al is not properly established. A particular deviation from standard networks is the notion of channel noise. I think that the framework can still be relevant for generative priors by considering it as the problem of inverting a network or finding the latent code. This would require some changes in the paper and its exposition.

---

> ### Author Response · Authors · 2022-08-01
> **Response to R4-xNsM**
>
> Thank you for carefully reviewing our work. In response to these questions:
>
> - __Question on noise in Definition 3.3__: thank you for the question, pointing out a misprint in the definition. The noise in Definition 3.3 should not be $\sigma^2 = 1$, but a generic $\sigma^2 \geq 0$ which includes $\sigma^2=0$. Indeed, our proof holds for generic noise including zero, and we show empirics for this case in Figure 5, where the signal prior is a noiseless ReLU generative prior. We completely agree that generative models usually consider activations with zero noise and this is included in our results. We state our results in the form with noise simply because it is more general and includes the traditional noiseless case. We will add a comment connecting back to the noiseless case that is more commonly considered.
>
> - __Mixing the notions of generative priors and multilayer measurement processes__: we agree that our model differs slightly from the model considered by Bora et al. 2017. In their model, the goal is to estimate the output $s$ of a generative model $s = G_\theta(x)$, while in our model, the goal is to `invert the generative network' by estimating an input $x$ such that $y = AG_\theta(x)$. However, in the case when each layer of $G_\theta(x)$ has a deterministic channel function (for example, $\sigma^2 = 0$ at each layer $1 \leq l \leq L-1$), estimating $x$ is sufficient because one can deterministically compute the corresponding $s = G_\theta(x)$. We will comment on this subtle difference and highlight how our multilayer signal model generalizes the setting of Bora et al. 2017, so that our results are indeed applicable to generative CS.
>     We plan to edit the discussion following equation (1) to clarify that our problem can be seen both as multilayer signal recovery and as generative CS (via inverting a generative model whose last layer implicitly contains the sensing matrix).
> - __Introductory discussion of Manoel et al. 2017__: we will update our discussion of Manoel et al. 2017 on Page 2 to refer to unstructured matrices instead of strictly dense Gaussian matrices. While Manoel et al. 2017 don’t require distributional assumptions on the sensing matrix coordinates, it is not known in general what conditions are necessary for rigorous state evolution to hold, so it could be misleading for us to refer to generic unstructured matrices in the context of proving state evolution. We emphasize dense Gaussian matrices in our work because they are a good representative comparison to Gaussian MCC matrices, with rigorous state evolution in both cases.
>
> - __Last paragraph of related work__: we agree that the discussion of stable training is unnecessary and potentially confusing. The main point of this paragraph is to highlight practical architectural features of the generative CNN models that our idealized setting is intended to capture. We will edit our discussion to focus only on architectural aspects of generative neural nets for signal recovery.
>
> - __Citations__: We will cite Rezende et al 2015, Dinh et al. 2015, before discussing GLOW/RealNVP and their application to generative signal recovery. We will change our discussion to indicate that Bora et al. 2017 do not only study a DC-GAN architecture, but also a dense MLP VAE.

---

> > ### Comment · Reviewer_xNsM · 2022-08-08
> > **Response to the authors**
> >
> > I would like to thank the authors for their answers, particularly the clarification regarding the channel noise. This resolves my concern. I appreciate if the authors can reflect on the assumption of random weights and whether (or how) the obtained state evolution could still bring insights for the trained networks. This is a valuable discussion for the main paper.
> >
> > For now, I increased my score of the paper.

---

> > > ### Author Response · Authors · 2022-08-09
> > > **Insights for trained networks**
> > >
> > > Concerning the insights our work can bring for trained networks: we will add our thoughts in the discussion of the paper. There are several trained settings for which the same state evolution could apply and that we are currently investigating. One possibility is to study the weight matrices of neural networks in the lazy training regime, in which the weights stay close to their initial values throughout training. AMP methods can also be modified to include partially deterministic structure in their sensing matrices and better represent learned models. For example, as we discuss in Appendix D and briefly in the body, our proof can easily be extended to study MCC matrices whose convolutional filters have non-isotropic variance, which is a simple model for convolutions with partially deterministic structure. As another example, [Gabrié et al. 2018](https://papers.nips.cc/paper/2018/hash/6d0f846348a856321729a2f36734d1a7-Abstract.html) propose an AMP-based method to compute mutual information between the activations at each layer of a deep random neural network. They observe in Section 3 that their AMP-based predictions hold for a linear model whose sensing matrices have random singular vectors and learned spectrum. As stated in the last sentence of our paper, trained settings for which the simple Gaussian state evolution does not apply anymore are even more interesting venue for future works.

---

### Author Response · Authors · 2022-08-03
**Updates to the text**

Based on feedback from the reviewers, we have updated the text in the body of our paper and reuploaded a new version. For the reviewers' convenience, we also provide a version with significant edits highlighted in blue:

[https://anonymous.4open.science/r/conv-ml-amp-draft-6BDA](https://anonymous.4open.science/r/conv-ml-amp-draft-6BDA)

Please keep in mind that we are still planning to modify our experiments by adding empirical evaluations of the corresponding dense Gaussian model (see response to R1-PaFY, 'Lack of numerical simulations').

We thank the reviewers for their consideration of our work.

---

### Meta-Review · Area_Chair_sFfX · 2022-08-26

**Recommendation:** Accept
**Confidence:** Certain

**Metareview:**

This paper considers finding an input vector from multi-layer noisy measurements. This can alternatively be thought of as finding the latent code of generative models. The authors analyze the state evolution of a multi-layer multi-layer approximate message passing algorithm. The main technical idea is relating random convolutional layers to Gaussian ones using permutation matrices and then utilizing a connection with spatially coupled matrices in coding. The authors also provide numerical evidence. Overall all the reviewers were positive but did raise some concerns about the model bing not realistic since the convolutional layers are not trained. I agree with the assessment of the reviewers. I think the paper is interesting and the connections and theoretical results are nice. Therefore I recommend acceptance. However, I do have some concerns about the model studied. I also have some concerns about the theoretical analysis as it is sometimes difficult to differentiate what is fully rigorous and what is based on statistical physics conjectures. In your final manuscript please clarify which parts are fully rigorous (perhaps all) and which parts rely on conjectures that have not been fully proved.


**Award:**

No

---

### Decision · Program_Chairs · 2022-09-14

Accept